# End-to-End Learning for Fair Multiobjective Optimization Under Uncertainty

**My H Dinh**[1] **James Kotary**[1] **Ferdinando Fioretto**[1]

[1]Department of Computer Science, University of Virginia, Charlottesville, Virginia, USA

## Abstract

Many decision processes in artificial intelligence and operations research are modeled by parametric optimization problems whose defining parameters are unknown and must be inferred from observable data. The Predict-Then-Optimize (PtO) paradigm in machine learning aims to maximize downstream decision quality by training the parametric inference model end-to-end with the subsequent constrained optimization. This requires backpropagation through the optimization problem using approximation techniques specific to the problem's form, especially for nondifferentiable linear and mixed-integer programs. This paper extends the PtO methodology to optimization problems with nondifferentiable Ordered Weighted Averaging (OWA) objectives, known for their ability to ensure properties of fairness and robustness in decision models. Through a collection of training techniques and proposed application settings, it shows how the optimization of OWA functions can be effectively integrated with parametric prediction for fair and robust optimization under uncertainty.

## 1 INTRODUCTION

The *Predict-Then-Optimize* (PtO) framework models decision-making processes as optimization problems with unspecified parameters $c$, which must be estimated by a machine learning (ML) model, given correlated features $z$. An estimation of $c$ completes the problem's specification, whose solution defines a mapping:

$$x^\star(c) = \underset{x \in \mathcal{S}}{\mathrm{argmax}} \; f(x, c) \tag{1}$$

The goal is to learn a model $\hat{c} = \mathcal{M}_\theta(z)$ from observable features $z$, such that the objective value $f(x^\star(\hat{c}), c)$ under ground-truth parameters $c$ is maximized on average.

This setting is common to many real-world applications requiring decision-making under uncertainty, such as planning the fastest route through a city with unknown traffic delays, or determining optimal power generation schedules based on demand forcasts. A classic example is the Markowitz portfolio problem, wherein the optimization model (1) may regard $f$ as the total return due to asset allocations $x$ under predicted prices $c$, while $\mathcal{S}$ includes constraints on price covariance as a measure of risk Markowitz [1991]. Modern approaches are based on *end-to-end learning*, and train $\hat{c} = \mathcal{M}_\theta(z)$ to maximize $f(x^\star(\hat{c}), c)$ directly as loss function. This requires backpropagation through $x^\star(\hat{c})$, which is especially challenging when (1) defines a nondifferentiable mapping, as further elaborated in Section 2.

Within this context, optimizing multiple objectives becomes crucial, requiring a balance of competing goals. This is especially important when the objectives need to be optimized fairly, a common requirement in engineering settings such as energy systems [Terlouw et al., 2019], urban planning [Salas and Yepes, 2020], and multi-objective portfolio optimization [Iancu and Trichakis, 2014, Chen and Zhou, 2022]. A prevalent approach in this setting is the optimization of a scalar aggregation of all objectives using Ordered Weighted Averaging (OWA) [Yager, 1993]. This approach results in Pareto-optimal solutions that fairly balance the values of each objective. However, employing optimization of an OWA objective in Predict-Then-Optimize is challenged due to its nondifferentiability, which prevents backpropagation of its constrained optimization mapping $x^\star(c)$ within machine learning models trained by gradient descent. To the best of our knowledge, no prior PtO models encounter a non-differentiable objective, making this challenge novel.

This paper aims to address this challenge and facilitate the integration of learning and optimization for novel applications such as fair learning-to-rank models based on OWA optimization of rankings and Markowitz prediction models based on multi-scenario portfolio optimization. By leveraging modern techniques in OWA optimization and Predict-Then-Optimize (PtO) learning, this paper shows how the

*Accepted for the 40th Conference on Uncertainty in Artificial Intelligence* (UAI 2024).

optimization of OWA functions can be effectively backpropagated in machine learning models, enabling end-to-end trainable prediction and decision models for applications requiring fair and robust decision-making under uncertainty.

**Contributions.** In particular, the paper makes the following contributions: **(1)** It proposes novel techniques for differentiating OWA optimization models with respect to their uncertain parameters, allowing their integration in end-to-end trainable ML models. **(2)** It is the first to show how loss functions based on OWA aggregation can be effectively used for supervising such end-to-end training. **(3)** Based on these contributions, it proposes several effective modeling strategies for combining parametric prediction with OWA optimization and evaluates them in novel application settings where optimal decisions must be fair or robust to multiple uncertain objective criteria. The experiments conducted serve to underscore the practical significance of integrating predictive modeling with OWA optimization, yielding promising results across diverse application settings.

## 2 PRELIMINARIES

Prior to discussing the paper's contribution, this section provides an overview of the concepts of optimizing OWA functions and implementing end-to-end training methods for both prediction and optimization.

### 2.1 OWA AND ITS OPTIMIZATION

The *Ordered Weighted Average* (OWA) operator [Yager, 1993] is a class of functions used for aggregating multiple independent values in settings requiring multicriteria evaluation and comparison [Yager and Kacprzyk, 2012]. Let $\mathbf{y} \in \mathbb{R}^m$ be a vector of $m$ distinct criteria, and $\tau : \mathbb{R}^m \to \mathbb{R}^m$ be the sorting map for which $\tau(\mathbf{y}) \in \mathbb{R}^m$ holds the elements of $\mathbf{y}$ in increasing order. Then for any $\mathbf{w}$ satisfying $\{\mathbf{w} \in \mathbb{R}^m : \sum_i w_i = 1, \mathbf{w} \geq 0\}$, the OWA aggregation with weights $\mathbf{w}$ is defined as a linear functional on $\tau(\mathbf{y})$:

$$\text{OWA}_{\mathbf{w}}(\mathbf{y}) = \mathbf{w}^T \tau(\mathbf{y}), \qquad (2)$$

which is piecewise-linear in $\mathbf{y}$ [Ogryczak and Śliwiński, 2003].

**Fair OWA.** This paper focuses on a specific instance of OWA, commonly known as *Fair OWA* Ogryczak et al. [2014], characterized by weights arranged in descending order: $w_1 > w_2 \ldots > w_n > 0$. Note that with monotonic weights, Fair OWA is also concave. Fair OWA objectives are increasingly popular in optimization as fairness gains attention in decision-making processes.

The following three properties of Fair OWA functions are crucial for their use in fairly optimizing multiple objectives: **(1)** *Impartiality* ensures that Fair OWA treats all criteria equally. This means that for any permutation $\sigma \in \mathcal{P}_m$,

where $\mathcal{P}_m$ is the set of all permutations of $[1, \ldots, m]$, the OWA aggregation with weights $\mathbf{w}$ yields the same result for any permutation of the input vector $\mathbf{y}$. **(2)** *Equitability* guarantees that marginal transfers from a criterion with a higher value to one with a lower value increase the OWA aggregated value. This condition holds that $\text{OWA}_{\mathbf{w}}(\mathbf{y}_\epsilon) > \text{OWA}_{\mathbf{w}}(\mathbf{y})$, where $\mathbf{y}_\epsilon = \mathbf{y}$ except at positions $i$ and $j$ where $(\mathbf{y}_\epsilon)_i = \mathbf{y}_i - \epsilon$ and $(\mathbf{y}_\epsilon)_j = \mathbf{y}_j + \epsilon$, assuming $y_i > y_j + \epsilon$. **(3)** *Monotonicity* ensures that $\text{OWA}_{\mathbf{w}}(\mathbf{y})$ is an increasing function of each element of $\mathbf{y}$. This property implies that solutions optimizing the OWA objectives (2) are Pareto Efficient solutions of the underlying multiobjective problem, thus no single criteria can be raised without reducing another Ogryczak and Śliwiński [2003]. This aspect is crucial in optimization, where Pareto-efficient solutions are always preferred over those that do not possess this attribute. Taken together, these properties define a notion of fairness in optimal solutions known as *equitable efficiency* Ogryczak and Śliwiński [2003]. Intuitively, OWA objectives lead to fair optimal solutions by always assigning the highest weights of $\mathbf{w}$ to the objective criteria in order of lowest current value.

### 2.2 PREDICT-THEN-OPTIMIZE LEARNING

The problem setting of this paper can be viewed within the framework of Predict-Then-Optimize. In general, a parametric optimization problem (1) models an optimal decision $\mathbf{x}^\star(\mathbf{c})$ with respect to unknown parameters $\mathbf{c}$ drawn from a distribution $\mathbf{c} \sim \mathcal{C}$. Although the true value of $\mathbf{c}$ is unknown, correlated *feature* values $\mathbf{z} \sim \mathcal{Z}$ can be observed. The goal is to learn a predictive model $\mathcal{M}_\theta : \mathcal{Z} \to \mathcal{C}$ from features $\mathbf{z}$ to estimate problem parameters $\hat{\mathbf{c}} = \mathcal{M}_\theta(\mathbf{z})$. This estimation aims to maximize the empirical objective value of the resulting solution under ground-truth parameters. That is,

$$\underset{\theta}{\arg\max} \quad \mathbb{E}_{(\mathbf{z}, \mathbf{c}) \sim \Omega} \; f\left(\mathbf{x}^\star(\mathcal{M}_\theta(\mathbf{z})), \mathbf{c}\right), \qquad (3)$$

where $\Omega$ represents the joint distribution between $\mathcal{Z}$ and $\mathcal{C}$.

The above training goal is often best realized by maximizing empirical *Decision Quality* as a loss function Mandi et al. [2023], defined as:

$$\mathcal{L}_{DQ}(\hat{\mathbf{c}}, \mathbf{c}) = f\left(\mathbf{x}^\star(\hat{\mathbf{c}}), \mathbf{c}\right). \qquad (4)$$

Gradient descent training of (3) with $\mathcal{L}_{DQ}$ requires a model of gradient $\frac{\partial \mathcal{L}_{DQ}}{\partial \hat{\mathbf{c}}}$, either directly or through chain-rule composition $\frac{\partial \mathcal{L}_{DQ}}{\partial \hat{\mathbf{c}}} = \frac{\partial \mathbf{x}^\star(\hat{\mathbf{c}})}{\partial \hat{\mathbf{c}}} \cdot \frac{\partial \mathcal{L}_{DQ}}{\partial \mathbf{x}^\star}$. Here, left-multiplication by the Jacobian is equivalent to backpropagation through the optimization mapping $\mathbf{x}^\star$. When $\mathbf{x}^\star$ is not differentiable, as in the case of OWA optimizations, smooth approximations are required, such as those developed in the next section.

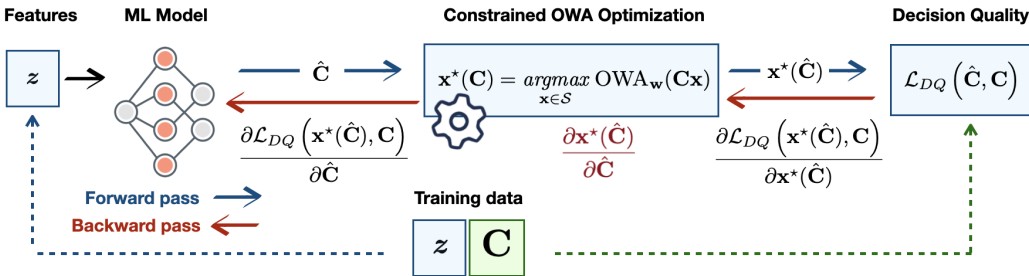

Figure 1: Predict-Then-Optimize for OWA Optimization.

# 3 END-TO-END LEARNING WITH FAIR OWA OPTIMIZATION

This paper's proposed methodology and settings focus on the scenarios where the objective function $f$ consists of an ordered weighted average of $m$ linear objective functions, with each function parametrized by one row of a matrix $C \in \mathbb{R}^{m \times n}$, so that $f(x, C) = \text{OWA}_w(Cx)$ and

$$x^\star(C) = \underset{x \in \mathcal{S}}{\text{argmax}} \ \text{OWA}_w(Cx). \quad (5)$$

Note that the methodology of this paper naturally extends to cases where the OWA objective above is combined with additional smooth objective terms. For simplicity, the exposition primarily focuses on the pure OWA objective as shown in equation (5), wherever applicable.

The goal is to learn a prediction model $\hat{C} = \mathcal{M}_\theta(z)$ that maximizes decision quality through gradient descent on problem (3), which requires obtaining its gradients w.r.t. $\hat{C}$:

$$\frac{\partial \mathcal{L}_{DQ}(\hat{C}, C)}{\partial \hat{C}} = \underbrace{\frac{\partial x^\star}{\partial \hat{C}}}_{J} \cdot \underbrace{\frac{\partial \text{OWA}_w(Cx^\star)}{\partial x^\star}}_{g}, \quad (6)$$

where $x^\star$ is evaluated at $\hat{C}$. The primary strategy for modeling this overall gradient involves initially determining the OWA function's gradient $g$, followed by computing the product $Jg$ by backpropagation of $g$ through $x^\star$.

While nondifferentiable, the class of OWA functions is *subdifferentiable*, with subgradients as follows:

$$\frac{\partial}{\partial y} \text{OWA}_w(y) = w_{(\sigma^{-1})} \quad (7)$$

where $\sigma$ are the sorting indices on $y$ [Do and Usunier, 2022]. Based on this formula, computing an overall subgradient $g = \partial/\partial x \ \text{OWA}_w(Cx)$ is a routine application of the chain rule (via automatic differentiation). The use of subgradients (7) in training ML models has been previously explored in the context of reinforcement learning Siddique et al. [2020]. This work also leverages subgradients to incorporate the fairness aspect of OWA optimization into end-to-end learning. A schematic illustration highlighting the forward and

backward steps required for this process is provided in Figure 1.

As outlined next, the main technical contribution of the paper is to propose differentiable models of OWA optimization (5), through which backpropagation of $g$ can effectively approximate the decision quality gradient $Jg$ for end-to-end training of (3). The following sections propose alternative models of differentiable OWA optimization, each taylored to address problem-specific technical challenges.

First, Section 4 demonstrates how the OWA optimization (5) with continuous variables can be effectively smoothed to yield differentiable approximations that can be backpropagated in end-to-end training (3). Next, Section 5 focuses on a special form of optimization mapping with a nonparametric OWA term as an additional parametric objective term, showing how backpropagation can be implemented using only a blackbox solver for the underlying problem, without smoothing. Finally, Section 6 outlines a method involving surrogate solvers for cases where OWA-aggregation of objectives makes optimization too difficult to solve directly.

# 4 DIFFERENTIABLE APPROXIMATE OWA OPTIMIZATION

This section develops two alternative differentiable approximations of the OWA optimization mapping (5). Prior works [Wilder et al., 2019b, Amos et al., 2019] show that when an optimization mapping (1) is discontinuous, as is the case when $f$ and $\mathcal{S}$ define a linear program (LP), differentiable approximations to (1) can be formed by regularization of its objective by smooth functions. Section 4.1 will demonstrate how linear programming models of OWA optimization can be combined with smoothing techniques for LP, yielding effective differentiable approximations of (5).

However, this model becomes computationally intractable for more than a few criteria $m$. An efficient alternative is proposed in Section 4.2, where the mapping (5) is made differentiable by replacing the OWA objective with its smooth Moreau envelope approximation. To the best of the author's knowledge, this is the first time that objective smoothing

via the Moreau envelope is used (and shown be an effective technique) for approximating nondifferentiable optimization programs in end-to-end learning. As approximations of the true mapping (5), both smoothed models are used employed in training and replaced by (5) at test time, similarly to a softmax layer in classification.

## 4.1 OWA LP WITH QUADRATIC SMOOTHING

The mainstay approach to solve problem (5) when $x \in \mathcal{S}$ is linear is to transform the problem into a linear program without OWA functions, and solve it with a simplex method [Ogryczak and Śliwiński, 2003]. Our first approach to differentiable OWA optimization combines this transformation with the smoothing technique of Wilder et al. [2019b], which forms differentiable approximations to linear programs

$$x^\star(c) = \text{argmax}_{Ax \leq b} \; c^T x \quad (8)$$

by adding a scaled euclidean norm term $\epsilon \|x\|^2$ to the objective function, resulting in a continuous mapping $x^\star(c) = \text{argmax}_{Ax \leq b} \; c^T x + \epsilon \|x\|^2$, a quadratic program (QP) which can be differentiated implicitly via its KKT conditions as in [Amos and Kolter, 2017].

We adapt a version of this technique to OWA optimization (5) by first forming an equivalent LP problem. It is observed in [Ogryczak and Śliwiński, 2003] that $\text{OWA}_w$ can be expressed as the minimum weighted average among all permutations of the OWA weights $w$:

$$OWA_{\mathbf{w}}(\mathbf{r}) = \max_z \; z \quad s.t. \quad z \leq \mathbf{w}_\sigma \cdot \mathbf{r}, \quad \forall \sigma \in \mathcal{P}, \quad (9)$$

which allows the OWA optimization (5) to be expressed as

$$x^\star(C) = \text{argmax}_{x \in \mathcal{S}, y, z} \; z \quad (10a)$$
$$\text{s.t.:} \quad y = Cx \quad (10b)$$
$$z \leq w_\tau \cdot y \quad \forall \tau \in \mathcal{P}_m. \quad (10c)$$

When the constraints $x \in \mathcal{S}$ are linear, problem (10) is a LP. However, its constraints (10c) grow factorially as $m!$, where $m$ is the number of individual objective criteria aggregated by OWA. Smoothing by the scaled norm of joint variables $x, y, z$ leads to a differentiable QP approximation, viable when $m$ is small. This optimization can be solved and differentiated using techniques from Amos and Kolter [2017] or a generic differentiable optimization solver such as Agrawal et al. [2019a]:

$$x^\star(C) = \underset{x \in \mathcal{S}, y, z}{\text{argmax}} \; z + \epsilon \left( \|x\|_2^2 + \|y\|_2^2 + z^2 \right) \quad (11a)$$
$$\text{subject to:} \quad (10b), (10c). \quad (11b)$$

While problem (10) does not fit the exact form (8) due to its parameterized constraints (10b), the need for quadratic smoothing (11a) is illustrated experimentally in Section 7.1.1. The main *disadvantage* of this method is poor scalability in the number of criteria $m$, due to constraints (10c).

Another disadvantage is that the transformed QP is much harder to solve than its original associated LP problems, since quadratic smoothing increases the difficulty of an OWA-equivalent LP problem. These drawbacks motivate the next smoothing method, which yields a tractable optimization problem by replacing the OWA objective itself with a smooth approximation.

## 4.2 MOREAU ENVELOPE SMOOTHING

In light of the efficiency challenges faced by (11), we propose an alternative smoothing technique to form more scalable differentiable approximations of the optimization mapping (5). Instead of adding a quadratic term as in (11), we replace the piecewise linear function $\text{OWA}_w$ in (5) with its Moreau envelope, defined for a convex function $f$ as:

$$f^\beta(\mathbf{x}) = \min_{\mathbf{v}} \; f(\mathbf{v}) + \frac{1}{2\beta} \|\mathbf{v} - \mathbf{x}\|^2. \quad (12)$$

Moreau envelopes of concave functions are defined analogously. Compared to its underlying function $f$, the Moreau envelope is $\frac{1}{\beta}$ smooth while sharing the same optima [Beck, 2017]. The Moreau envelope-smoothed OWA optimization problem is

$$x^\star(C) = \text{argmax}_{x \in \mathcal{S}} \; \text{OWA}_w^\beta(Cx). \quad (13)$$

With its smooth objective function, problem (13) can be solved by gradient-based optimization methods, such as projected gradient descent, or more likely a Frank-Wolfe method if $x \in \mathcal{S}$ is linear (see Section 7.1.1). Additionally, it can be effectively backpropagated in end-to-end learning.

Backpropagation of (13) is nontrivial since its objective function lacks a closed form. To proceed, we first note from [Do and Usunier, 2022] that the gradient of the Moreau envelope $\text{OWA}_w^\beta$ is equal to a Euclidean projection:

$$\frac{\partial}{\partial x} \text{OWA}_w^\beta(\mathbf{x}) = \text{proj}_{\mathcal{C}(\tilde{w})} \left( \frac{x}{\beta} \right), \quad (14)$$

where $\tilde{w} = -(w_m, \dots, w_1)$ and the permutahedron $\mathcal{C}(\tilde{w})$ is the convex hull of all permutations of $\tilde{w}$. It's further shown in [Blondel et al., 2020] how such a projection can be computed and differentiated in $\mathcal{O}(m \log m)$ time using isotonic regression. To leverage the differentiable gradient function (32) for backpropagation of the smoothed optimization (13), we model its Jacobian by differentiating the fixed-point conditions of a gradient-based solver.

Letting $\mathcal{U}(x, C) = proj_{\mathcal{S}}(x - \alpha \cdot \frac{\partial}{\partial x} \text{OWA}_w^\beta(x, C))$, a projected gradient descent step on (13) is $x^{k+1} = \mathcal{U}(x^k, C)$. Differentiating the fixed-point conditions of convergence where $x^k = x^{k+1} = x^\star$, and rearranging terms yields a linear system for $\frac{\partial x^\star}{\partial C}$:

$$\left(I - \underbrace{\frac{\partial \mathcal{U}(\boldsymbol{x}^\star, \boldsymbol{C})}{\partial \boldsymbol{x}^\star}}_{\boldsymbol{\Phi}}\right) \frac{\partial \boldsymbol{x}^\star}{\partial \boldsymbol{C}} = \underbrace{\frac{\partial \mathcal{U}(\boldsymbol{x}^\star, \boldsymbol{C})}{\partial \boldsymbol{C}}}_{\boldsymbol{\Psi}} \qquad (15)$$

The partial Jacobian matrices $\boldsymbol{\Phi}$ and $\boldsymbol{\Psi}$ above can be found given a differentiable implementation of $\mathcal{U}$. This is achieved by computing the inner gradient $\frac{\partial}{\partial \boldsymbol{x}} \mathrm{OWA}_{\boldsymbol{w}}^{\beta}(\boldsymbol{x}, \boldsymbol{C})$ via the differentiable permutahedral projection (32), and solving the outer projection mapping $\mathrm{proj}_{\mathcal{S}}$ using a generic differentiable solver such as `cvxpy` [Agrawal et al., 2019a]. As such, applying $\mathcal{U}$ at a precomputed solution $\boldsymbol{x}^\star(\boldsymbol{C})$ allows $\boldsymbol{\Phi}$ and $\boldsymbol{\Psi}$ to be extracted in PyTorch, in order to solve (15); this process is efficiently implemented via the `fold-opt` library [Kotary et al., 2023].

# 5 BLACKBOX METHODS FOR NONPARAMETRIC OWA OBJECTIVE

This section proposes a special class of techniques for cases where the OWA term of an objective function is specified with *known* coefficients $\boldsymbol{B} \in \mathbb{R}^{m \times n}$, and uncertainty lies instead in an additional parametrized *linear* objective term:

$$\boldsymbol{x}^\star(\boldsymbol{c}) = \mathrm{argmax}_{\boldsymbol{x} \in \mathcal{S}} \ \boldsymbol{c}^T \boldsymbol{x} + \lambda \mathrm{OWA}_{\boldsymbol{w}}(\boldsymbol{B}\boldsymbol{x}). \qquad (16)$$

This form is taken by the optimization mapping within the fair learning to rank model proposed in Section 7.2. By employing the reformulation (10), (16) transforms into:

$$(\boldsymbol{x}^\star, \boldsymbol{y}^\star, z^\star)(\boldsymbol{c}) = \mathrm{argmax}_{\boldsymbol{x} \in \mathcal{S}, \boldsymbol{y}, z} \ \boldsymbol{c}^T \boldsymbol{x} + \lambda z \qquad (17\mathrm{a})$$

$$\text{s.t.:} \ \boldsymbol{y} = \boldsymbol{B}\boldsymbol{x} \qquad (17\mathrm{b})$$

$$z \le \boldsymbol{w}_\tau \cdot \boldsymbol{y} \quad \forall \tau \in \mathcal{P}, \qquad (17\mathrm{c})$$

which as discussed in Subsection 4.1 grows intractable with increasing $m$ since the constraints (17c) number $(m!)$.

We observe that the optimization problem 17 fits the particular form $\boldsymbol{v}^\star(\boldsymbol{\gamma}) = \mathrm{argmax}_{\boldsymbol{v} \in \mathcal{C}} \ \boldsymbol{\gamma}^T \boldsymbol{v}$, as treated in several works [Elmachtoub and Grigas, 2021, Berthet et al., 2020, Pogančić et al., 2020], where an uncertain *linear* objective is paired with *nonparametric* constraints. These works propose differentiable solvers based on *blackbox* solvers of the underlying optimization problem, without smoothing. This is generally accomplished by modeling the gradient as a combination of solutions induced by perturbed input parameters. As shown next, this allows us to compute a gradient formula for (17) without solving it directly. Instead, we use a black-box solver for the underlying problem (16).

We illustrate the idea using the "Smart Predict-Then-Optimize" scheme [Elmachtoub and Grigas, 2021], which trains to maximize $\mathcal{L}_{DQ}$ by equivalently minimizing the suboptimality (called *regret*) via a convex subdifferentiable upper bounding function named $\mathcal{L}_{SPO+}$. By construction, it admits a formula for subgradients directly with respect to $\hat{\boldsymbol{c}}$:

$$\partial/\partial \hat{\boldsymbol{\gamma}} \ L_{\mathrm{SPO+}}(\hat{\boldsymbol{\gamma}}, \boldsymbol{\gamma}) = \boldsymbol{v}^\star(2\hat{\boldsymbol{\gamma}} - \boldsymbol{\gamma}) - \boldsymbol{v}^\star(\boldsymbol{\gamma}). \qquad (18)$$

Given any efficient method that provides optimal solutions $\boldsymbol{x}^\star(\boldsymbol{C})$ to the OWA optimization (16), the auxiliary variables of problem (17) can be recovered as $\boldsymbol{y}^\star = \boldsymbol{C}\boldsymbol{x}^\star$ and $z^\star = \mathrm{OWA}_{\boldsymbol{w}}(\boldsymbol{y}^\star)$. Defining the variables $\boldsymbol{v}^\star = (\boldsymbol{x}^\star, \boldsymbol{y}^\star, z^\star)$ and noting that $\boldsymbol{\gamma} = (\boldsymbol{c}, \boldsymbol{0}, \lambda)$ in problem (17), its SPO+ loss subgradient can be now computed directly using formula (18). This approach leverages the problem form (17) to derive a *backpropagation model* while *avoiding its direct solution* as a linear program. Section 7.1.1 will show how this can be combined with an efficient Frank-Wolfe solution of (16) to design a scalable fair learning-to-rank model.

# 6 DIFFERENTIABLE SURROGATE OPTIMIZATION MAPPINGS

OWA optimization problems (5) can be challenging to solve, even with modern methods, unless special problem-specific structures are exploited. In such cases, an alternative to the differentiable approximations of (5) proposed in Section 4 involves generating feasible candidate solutions $\boldsymbol{x}^\star \in \mathcal{S}$ from a simpler differentiable model without OWA objectives.

For example, a *linear* surrogate model proves useful when (5) represents fair OWA optimization of multiple objectives in a linear program (such as shortest path or bipartite matching) which depends on total unimodularity to maintain integral solutions $\boldsymbol{x}^\star \in \{0, 1\}^n$:

$$\boldsymbol{x}^\star(\boldsymbol{c}) = \mathrm{argmax}_{\boldsymbol{x} \in \mathcal{S}} \ \boldsymbol{c}^T \boldsymbol{x} \qquad (19)$$

As illustrated in Section 7.1.2 on a parametric shortest path problem, this surrogate approach is essential to avoid arising an intractable OWA mixed-integer program, since integrality of solutions is guaranteed only under linear objectives.

The main *disadvantage* inherent to the proposed surrogate models is that they do not directly approximate the true OWA problem (5). Thus, the learned model $\mathcal{M}_\theta(\boldsymbol{z}) = \hat{\boldsymbol{c}} \in \mathbb{R}^n$ does not fit the form prescribed in ((3),(5)) as written, and it cannot supply parametric estimates $\hat{\boldsymbol{C}} \in \mathbb{R}^{m \times n}$ to an external solver of problem (5). Despite this, using $\mathrm{OWA}_{\boldsymbol{w}}(\boldsymbol{C}\boldsymbol{x}(\hat{\boldsymbol{c}}))$ as a loss function trains the surrogate model to learn solutions to (5) with high decision quality.

# 7 EXPERIMENTS

This section leverages the differentiable elements introduced in Sections 4-6 to develop end-to-end trainable prediction and OWA optimization models. Three experimental settings are proposed for evaluation across two primary application

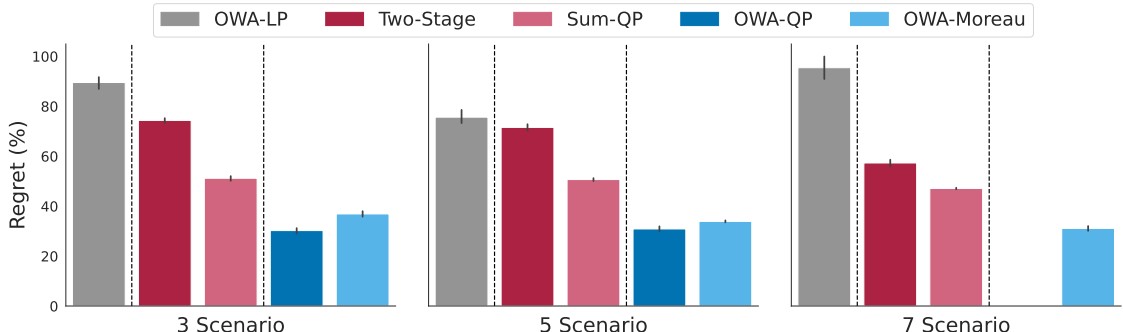

Figure 2: Percentage OWA regret (lower is better) on test set, on robust portfolio problem over 3,5,7 scenarios.

settings. The first application setting is *Fair Multiobjective Prediction and Optimization*, extending the Predict-Then-Optimize framework of Section 2.2 to scenarios where multiple uncertain objective functions must be jointly learned and fairly optimized through OWA aggregation, as defined in (5). Within this setting, *Robust Markowitz Portfolio Optimization* focuses on comparatively evaluating the differentiable approximations proposed in Section 4 against various baseline methods. Then, *Multi-Species Warcraft Shortest Path* serves as a case study demonstrating how a differentiable surrogate model can enable learning with OWA optimization of integer variables. The second application setting presents a *Fair Learning-to-Rank* model. Here, the OWA-aggregated objectives are known with certainty, corresponding to problem (16), as detailed in Section 5.

## 7.1 FAIR MULTIOBJECTIVE PREDICT-AND-OPTIMIZE

This setting employs a prediction model $\hat{C} = \mathcal{M}_\theta(z)$ to jointly estimate, from features $z$, the coefficients $C \in \mathbb{R}^{m \times n}$ of $m$ linear objectives, taken together as $Cx \in \mathbb{R}^m$. Its training goal is to maximize empirical decision quality with respect to their Fair OWA aggregation $f(x, C) = \text{OWA}_w(Cx)$:

$$\mathcal{L}_{DQ}(\hat{C}, C) = \text{OWA}_w\left(Cx^\star(\hat{C})\right). \quad (20)$$

Any descending OWA weights $w$ can be used to specify (20); we choose the squared *Gini indices* $w_j = \left(\frac{n-1+j}{n}\right)^2$.

**Evaluation.** In this section, each model is evaluated on the basis of its ability to train a model $\hat{C} = \mathcal{M}_\theta(z)$ to attain high decision quality (20) in terms of the OWA-aggregated objective. Results are reported in terms of the equivalent *regret* metric of suboptimality, whose minimimum value 0 corresponds to maximum decision quality:

$$regret(\hat{C}, C) = \text{OWA}_w^\star(C) - \text{OWA}_w\left(Cx^\star(\hat{C})\right) \quad (21)$$

where $\text{OWA}_w^\star(C)$ is the true optimal value of problem (5). This experiment is designed to evaluate the proposed

differentiable approximations (11) and (13) of Section 4; for reference they are named *OWA-QP* and *OWA-Moreau*.

**Baseline Models.** In addition to the newly proposed models, the evaluations presented in this section include two main baseline methods: **(1)** The *two-stage* method is the standard baseline for comparison against proposed methods for Predict-Then-Optimize training (3) [Mandi et al., 2023]. It trains the prediction model $\hat{C} = \mathcal{M}_\theta(z)$ by MSE regression, minimizing $\mathcal{L}_{TS}(\hat{C}, C) = \|\hat{C} - C\|^2$ without considering the downstream optimization model, which is employed only at test time. In addition, **(2)** the *unweighted sum* (*UWS*) of the objective criteria results in an LP mapping $x^\star(C) = \text{argmax}_{x \in \mathcal{S}} \mathbf{1}^T(Cx)$ which can be employed in end-to-end training by using quadratic smoothing [Wilder et al., 2019b] in 7.1.1 and blackbox differentiation [Pogančić et al., 2020] in 7.1.2; this baseline (denoted as Sum-QP) leverages end-to-end learning but without incorporating the OWA objective.

### 7.1.1 Differentiable OWA Optimization: Robust Markowitz Portfolio Problem

The classic Markowitz portfolio problem is concerned with constructing an optimal investment portfolio, given future returns $c \in \mathbb{R}^n$ on $n$ assets, which are unknown and predicted from exogenous data. A common formulation maximizes future returns subject to a risk limit, modeled as a quadratic covariance constraint. Define the set of valid fractional allocations $\Delta_n = \{x \in \mathbb{R}^n : \mathbf{1}^T x = 1, x \geq 0\}$, then :

$$x^\star(c) = \underset{x \in \Delta_n}{\text{argmax}} \quad c^T x \quad s.t.: \quad x^T \Sigma x \leq \delta. \quad (22)$$

where $\Sigma \in \mathbb{R}^{n \times n}$ are the price covariances over $n$ assets. The optimal portfolio allocation (22) as a function of future returns $c \in \mathbb{R}^n$ is differentiable using known methods [Agrawal et al., 2019a], and is commonly used in evaluation of Predict-Then-Optimize methods [Mandi et al., 2023].

An alternative approach to risk-aware portfolio optimization views risk in terms of robustness over alternative scenarios. In [Cajas, 2021], $m$ future price scenarios are modeled by a

matrix $C \in \mathbb{R}^{m \times n}$ whose $i^{th}$ row holds per-asset prices in the $i^{th}$ scenario. Thus an optimal allocation is modeled as

$$\boldsymbol{x}^{\star}(\boldsymbol{C}) = \underset{\boldsymbol{x} \in \Delta_n}{\arg\max} \quad \text{OWA}_{\boldsymbol{w}}(\boldsymbol{C}\boldsymbol{x}). \tag{23}$$

This experiment integrates robust portfolio optimization (23) end-to-end with per-scenario price prediction $\hat{C} = \mathcal{M}_\theta(\boldsymbol{z})$.

**Settings.** Historical prices of $n = 50$ assets are obtained from the Nasdaq online database [Nasdaq, 2022] years 2015-2019, and $N = 5000$ baseline asset price samples $\boldsymbol{c}_i$ are generated by adding Gaussian random noise to randomly drawn price vectors. Price scenarios are simulated as a matrix of multiplicative factors uniformly drawn as $\mathcal{U}(0.5, 1.5)^{m \times n}$, whose rows are multiplied elementwise with $\boldsymbol{c}_i$ to obtain $\boldsymbol{C}_i \in \mathbb{R}^{m \times n}$. While future asset prices can be predicted on the basis of various exogenous data including past prices or sentiment analysis, this experiment generates feature vectors $\boldsymbol{z}_i$ using a randomly generated nonlinear feature mapping as described in Appendix A. The experiment is replicated in three settings which assume $m = 3, 5$, and 7 scenarios.

The predictive model $\mathcal{M}_\theta$ is a feedforward neural network. At test time, $\mathcal{M}_\theta$ is evaluated over a test set for the distribution $(\boldsymbol{z}, \boldsymbol{C}) \in \Omega$, by passing its predictions to a projected subgradient solver of (23). Complete details in Appendix A.

**Results.** Figure 2 shows percent regret in the OWA objective attained on average over the test set (lower is better). The end-to-end trained unweighted sum baseline outperforms the two-stage approach. However, both *OWA-QP* and *OWA-Moreau* achieve substantially higher decision quality. While *OWA-QP* performs slightly better, it cannot scale past 5 scenarios, highlighting the importance of the proposed Moreau envelope smoothing technique (Section 4.2).

OWA-LP represents a baseline method where the OWA's equivalent linear program (LP) is used as a differentiable optimization without smoothing. For comparison, the grey bars indicate the results from a non-smoothed OWA LP (9) implemented with implicit differentiation in `cvxpylayers` [Agrawal et al., 2019a]. This comparison highlights the improvement in accuracy due to applying quadratic smoothing in OWA-QP. The poor performance of the OWA subgradient training under the non-smoothed OWA-LP demonstrates that the proposed approximations in Section 4 are indeed necessary for accurate training.

Runtimes of the smoothed models (11) and (13) are compared in Figure 6 (Appendix A.1). These results show that the Moreau envelope smoothing maintains low runtimes as $m$ increases, while the QP approximation suffers past $m = 5$ and causes memory overflow beyond $m = 6$.

### 7.1.2 Surrogate Learning for OWA Optimization: Multi-species Warcraft Shortest Path

This experiment illustrates how a surrogate model can facilitate end-to-end training (3) when the full OWA problem (5) is too challenging to solve directly. he Warcraft Shortest Path (WSP) dataset is commonly used for benchmarking Predict-Then-Optimize (PtO) methods [Pogančić et al., 2020, Berthet et al., 2020]. In this dataset, observable features $\boldsymbol{z}$ are RGB images of $12 \times 12$ tiled Warcraft maps. A character's movement speed varies based on the terrain type of each tile, and the goal is to predict the node-weighted shortest path from the top-left to the bottom-right corner, where nodes represent tiles and weights correspond to movement speeds.

This experiment is a multi-objective variation inspired by [Tang and Khalil, 2023], where multiple species have distinct node weights determined by their movement speeds on each terrain type. The objective is for all species to traverse each map together along a single path as quickly as possible. To achieve this, we aim to minimize their OWA-aggregated path lengths.

Noting that node weights can readily be converted to edge weights, the shortest path problem as a linear program reads

$$\boldsymbol{x}^{\star}(\boldsymbol{c}) = \arg\max_{\boldsymbol{A}\boldsymbol{x}=\boldsymbol{b}, \, 0 \le \boldsymbol{x} \le 1} \quad -\boldsymbol{c}^T \boldsymbol{x} \tag{24}$$

where $\boldsymbol{A}$ is a graph incidence matrix, $\boldsymbol{b}$ indicates source and sink nodes, and $\boldsymbol{c}$ holds the graph's edge weights. A classic result states that due to total unimodularity in $A$, solutions $\boldsymbol{x}^{\star}$ to (24) are guaranteed to take on integer values, so that $\boldsymbol{x}^{\star}(\boldsymbol{c}) \in \{0, 1\}^n$ form valid paths [Cormen et al., 2022].

Replacing the linear objective of (24) with an OWA aggregation over $\boldsymbol{C}\boldsymbol{x}$ (where rows of $\boldsymbol{C} \in \mathbb{R}^{m \times n}$ hold edge weights per species) breaks this property, so that additional integer constraints $\boldsymbol{x} \in \{0, 1\}^n$ are required, leading to an intractable OWA integer program:

$$\boldsymbol{x}^{\star}(\boldsymbol{C}) = \arg\max_{\boldsymbol{A}\boldsymbol{x}=\boldsymbol{b}, \, \boldsymbol{x} \in \{0,1\}^n} \quad \text{OWA}_{\boldsymbol{w}}(-\boldsymbol{C}\boldsymbol{x}) \tag{25}$$

Rather than training a predictor of $\hat{\boldsymbol{C}} \in \mathbb{R}^{m \times n}$ together with (25), we predict $\hat{\boldsymbol{c}} \in \mathbb{R}^n$ with (24) as a differentiable LP surrogate model using [Pogančić et al., 2020], along with the OWA aggregated path length $\text{OWA}_{\boldsymbol{w}}(-\boldsymbol{C}\boldsymbol{x}^{\star}(\hat{\boldsymbol{c}}))$ as a loss function. This ensures integrality of $\boldsymbol{x}^{\star}(\boldsymbol{c})$ while maintaining an efficient training procedure which requires only to solve (24) at each training iteration and at inference.

**Settings.** Three species' node weights are derived by reassigning the movement speeds of each terrain type in the WSP dataset: Humans are fastest on land, Naga on water, and Dwarves on rock, to generate ground-truth $\boldsymbol{C} \in \mathbb{R}^{m \times n}$. $\mathcal{M}_\theta$ is a ResNet18 CNN trained to map $12 \times 12$ tiled Warcraft maps to node weights of a shortest path problem (24). Blackbox differentiation [Pogančić et al., 2020] is used

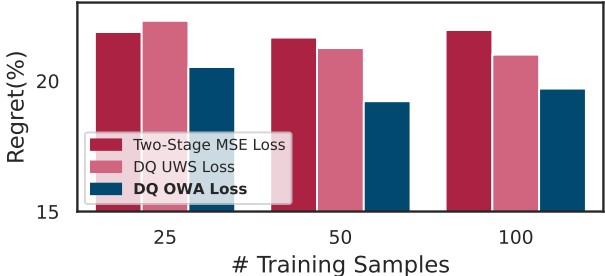

Figure 3: Multi-species OWA path length regret on WSP.

to backpropagate its solution by Dijkstra's method. See Appendix C for an example of the input feature data $\boldsymbol{z}$. ResNet18 is strong enough to enable competitive performance by the two-stage given enough data. Following [Tang and Khalil, 2023], we focus on the limited-data regime, where the advantage of end-to-end learning is best revealed, using 25, 50 or 100 training samples and 1000 for testing.

**Results.** Figure 3 records percentage regret due to two-stage and unweighted sum baseline models, along with the proposed differentiable LP surrogate trained under OWA loss. Notice how, in each case, the OWA-trained model shows a clear advantage in minimization of OWA regret.

Table 3 (Appendix C) shows the per-species regret in terms of path length, and reveals that OWA training significantly improves the highest path length among species, which is intuitive to provide fairness and the main contributor to the aggregated OWA value.

### 7.2 NONPARAMETRIC OWA WITH BLACKBOX SOLVER: FAIR LEARNING TO RANK

The final application setting studies the fair learning-to-rank problem, where a prediction model ranks $n$ web search results based on their relevance to a user query while ensuring fairness of exposure across protected groups within the search results. The proposed model learns relevance scores $\boldsymbol{c}$ end-to-end with a fair ranking optimization module:

$$\boldsymbol{\Pi}^{\star}(\boldsymbol{c}) = \mathrm{argmax}_{\boldsymbol{\Pi} \in \mathcal{B}} \ \ (1-\lambda) \cdot \boldsymbol{c}^T \boldsymbol{\Pi} \, \boldsymbol{b} + \lambda \cdot \mathrm{OWA}_{\boldsymbol{w}}(\mathcal{E}_G(\boldsymbol{\Pi})), \tag{26}$$

wherein $\mathcal{B}$ is the set of all bistochastic matrices, $\boldsymbol{\Pi} \in \mathbb{R}^{n \times n}$ represents a ranking policy whose $(i,j)^{th}$ element is the probability item $i$ takes position $j$ in the ranking, $\boldsymbol{c}$ measures relevance of each item to a user query, $\boldsymbol{b}$ are position bias factors measuring the exposure of each ranking position, and $\boldsymbol{c}^T \boldsymbol{\Pi} \, \boldsymbol{b}$ is the expected Discounted Cumulative Gain, a common measure of user utility. This primary objective is combined with OWA aggregation of the exposure vector $\mathcal{E}_G(\boldsymbol{\Pi})$, whose elements $\mathcal{E}_g(\boldsymbol{\Pi}) = \mathbf{1}_g^T \boldsymbol{\Pi} \, \boldsymbol{b}$ measure the exposures attained by each of several protected groups $g \in G$ where $\mathbf{1}_g$ hold binary indicators of item inclusion

in group $g$. The factor $\lambda$ controls a tradeoff between user utility and group fairness of exposure.

Since $\boldsymbol{b}$ and $\mathbf{1}_g$ in $\mathcal{E}_G(\boldsymbol{\Pi})$ are known and not modeled parametrically, the problem (26) is an instance of (16) and its SPO+ subgradient can be modeled as per Section 5. Solutions $\boldsymbol{\Pi}^{\star}(\boldsymbol{c})$ are obtained for any $\boldsymbol{c}$ by an adaptation of the Frank-Wolfe method with smoothing proposed in [Do and Usunier, 2022], as detailed in Appendix B.

**Settings.** A feedforward network $\mathcal{M}_\theta$ is trained to predict for $n$ items, given features $\boldsymbol{z}$, their relevance scores $\boldsymbol{c} \in \mathbb{R}^n$. The SPO+ training scheme of Section 5 is used to minimize regret in (26) due to error in $\hat{\boldsymbol{c}} = \mathcal{M}_\theta(\boldsymbol{z})$. The Microsoft Learning to Rank (MSLR) dataset is used, where $\boldsymbol{z}$ are features of items to be ranked and $\boldsymbol{c}$ are their relevance scores. Protected item groups are assigned as evenly spaced quantiles of its Quality Score feature. Each method is evaluated on the basis of mean utility $\boldsymbol{c}^T \boldsymbol{\Pi} \, \boldsymbol{b}$ and fairness violation $\frac{1}{|G|} \sum_{g \in G} \left| \frac{1}{n} \mathbf{1}^T \mathcal{E}_G(\boldsymbol{\Pi}) - \mathcal{E}_g(\boldsymbol{\Pi}) \right|$, and their relative tradeoffs over the full range of its fairness parameter.

**Baseline Models.** The model proposed in this section is called Smart OWA Optimization for Fair Ranking (SOFaiR). Selected baseline methods from the fair learning to rank domain include FULTR [Singh and Joachims, 2019], DELTR [Zehlike and Castillo, 2020], and SPOFR [Kotary et al., 2022], futher details are provided in Appendix B.

**Results.** Figure 9 shows that by enforcing fairness via an embedded optimization, SoFaiR achieves order-of-magnitude lower fairness violations than FULTR or DELTR, which rely on loss function penalties to drive down violations. However, it is Pareto-dominated over a small regime by those methods. Its fairness-utility tradeoff is comparable to SPOFR, which also uses constrained optimization. Notably though, SoFaiR demonstrates order-of-magnitude runtime advantages over SPOFR in Appendix B.

Figure 5 shows the analogous result over datasets with 3-7 protected groups. None of the baseline methods are equipped to handle multiple groups on this dataset, but SOFaiR accomodates more groups naturally by OWA optimization over their expected group exposures.

## 8 RELATED WORK

Modern approaches to the Predict-Then-Optimize setting, formalized in Section 2.2, typically maximize decision quality as a loss function, enabled by backpropagation through the mapping $\boldsymbol{c} \to \boldsymbol{x}^{\star}(\boldsymbol{c})$ defined by (1). When this mapping is differentiable, backpropagation can be performed using differentiable optimization libraries Amos and Kolter [2017], Agrawal et al. [2019a,b], Kotary et al. [2023].

However, many important classes of optimization are non-

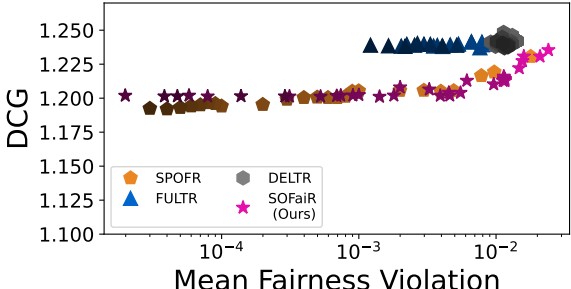

Figure 4: Fairness-utility tradeoffs on MSLR.

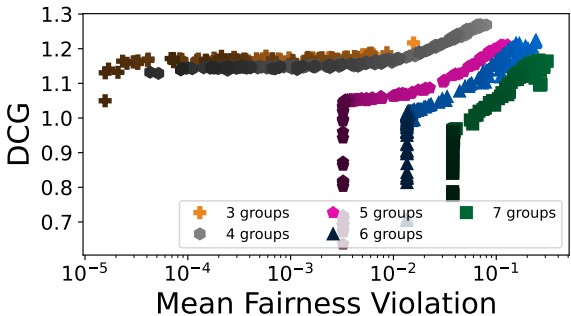

Figure 5: Fairness-utility tradeoffs on MSLR Multi-group.

differentiable, including linear and mixed-integer programs. Effective training techniques are typically based on forming continuous approximations of (1), whether by smoothing the objective function Amos et al. [2019], Wilder et al. [2019a], Mandi and Guns [2020], introducing random noise Berthet et al. [2020], Paulus et al. [2020], or estimation by finite differencing Pogančić et al. [2019]. This paper falls into that category, due to nondifferentiability of the OWA objective, requiring approximation of (1) by differentiable functions.

## 9 CONCLUSIONS

This work has presented a comprehensive methodology for integrating Fair OWA optimization seamlessly with predictive modeling within the Predict-then-Optimize paradigm. Our paper provides the tools to incorporate objective functions with robust fairness properties into integrated prediction and decision models. It contributes novel modeling techniques tailored to this context and illustrates how existing Predict-then-Optimize techniques can be adapted in nontrivial ways to maximize OWA effectiveness in this setting. Starting with innovative differentiable approximations to OWA programs, our proposed toolset includes specialized techniques to exploit problem-specific structures encountered in practical applications, such as nonparametric OWA objectives and totally unimodular constraints. These developments showcase the potential of Fair OWA optimization in data-driven decision-making, achieving results that were

previously unattainable for significant problems like robust resource allocation and fair learning-to-rank. We believe that this work could pave the way for the utilization of Fair OWA in learning pipelines, enabling a wide range of critical multi-optimization problems across various engineering domains.

## 10 ACKNOWLEDGEMENTS

This research is partially supported by NSF grants 2232054, 2133169, and NSF CAREER Award 2143706. The views and conclusions of this work are those of the authors only.

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

# Appendix

**My H Dinh**[1]  **James Kotary**[1]  **Ferdinando Fioretto**[1]

[1]Department of Computer Science, University of Virginia, Charlottesville, Virginia, USA

## A  PORTFOLIO OPTIMIZATION EXPERIMENT

### A.1  EFFICIENCY OF DIFFERENTIABLE OWA SOLVERS

Figure 6 depicts the running times of two differentiable optimization models applied to the Portfolio problem. It is evident that for the OWA-LP model, the running time scales factorially with the number of scenarios due to the number of constraints, while for the OWA-Moreau model, it scales linearly. It is noteworthy that the OWA-LP model cannot run with more than 7 scenarios due to memory constraints (requiring over 300GB+).

### A.2  EFFECT OF ADDING MSE LOSS

Figure 7 illustrates the impact of combining the Mean Squared Error loss $\mathcal{L}_{MSE}$ in a weighted combination with the decision quality loss $\mathcal{L}_{DQ}$. With the exception of OWA-LP, which exhibited instability, and Two-Stage, already trained with MSE Loss, the addition of MSE resulted in slight enhancements to the regret performance.

### A.3  MODELS AND HYPERPARAMETERS

A neural network (NN) with three shared hidden layers following by one separated hidden layer for each species is trained using Adam Optimizer and with a batch size of 64. The size of each shared layer is halved, the output dimension of the separated layer equal to the number of assets. Hyperparameters were selected as the best-performing on average among those listed in Table 1). Results for each hyperparameter setting are averaged over five random seeds. In the OWA-Moreau model, the forward pass is executed using projected gradient descent for 300, 500, and 750 iterations, respectively, for scenarios with 3, 5, and 7 inputs. The update step size is set to $\gamma = 0.02$.

Table 1: Hyperparameters

| Hyperparameter | Min | Max | Final Value | | | | | |
|---|---|---|---|---|---|---|---|---|
| | | | OWA-LP | Two-Stage | Sum-QP | OWA-QP | OWA-Moreau | Sur-QP |
| learning rate | $1e^{-3}$ | $1e^{-1}$ | $1e^{-2}$ | $5e^{-3}$ | $1e^{-2}$ | $1e^{-2}$ | $1e^{-2}$ | $1e^{-2}$ |
| smoothing parameter $\epsilon$ | 0.1 | 1.0 | N/A | N/A | 1.0 | 1.0 | N/A | 1.0 |
| smoothing parameter $\beta_0$ | 0.005 | 10.0 | N/A | N/A | N/A | N/A | 0.05 | N/A |
| MSE loss weight $\lambda$ | 0.1 | 0.5 | 0.4 | N/A | 0.3 | 0.4 | 0.1 | 0.3 |

*Accepted for the 40th Conference on Uncertainty in Artificial Intelligence* (UAI 2024).

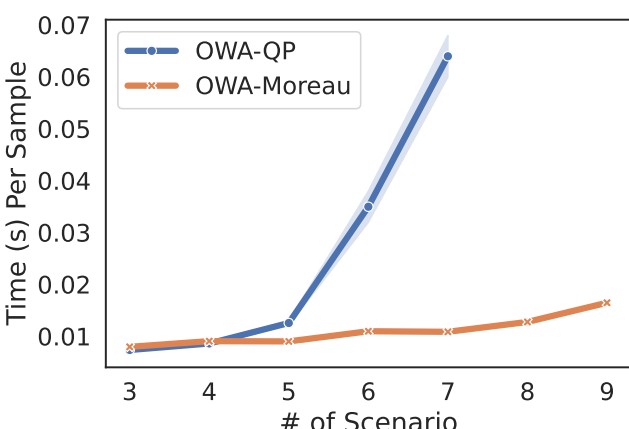

Figure 6: Average solving time of 2 smoothed OWA optimization models, on Robust Portfolio Optimization, over 1000 input samples. Missing datapoints past 7 scenarios are due to memory overflow as the QP model grows factorially.

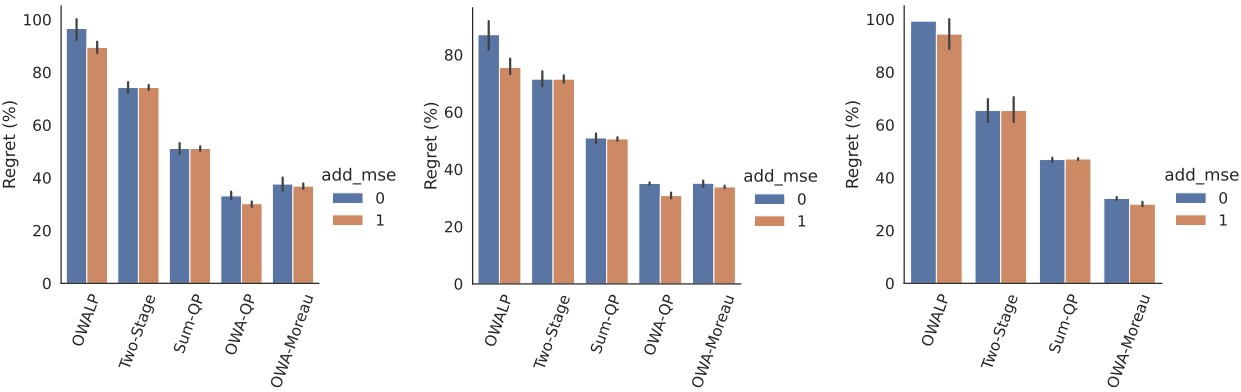

Figure 7: Effect of MSE Loss on differentiable optimization models. From left to right: 3, 5, 7 scenarios

## A.4 SOLUTION METHODS

The OWA portfolio optimization problem (23) is solved at test time, for each compared method, by projected subgradient descent using OWA subgradients (7) and an efficient projection onto the unit simplex $\Delta$ as in Martins and Astudillo [2016]:

$$\boldsymbol{x}^{k+1} = \text{proj}_\Delta \left( \boldsymbol{x}^k - \alpha \frac{\partial}{\partial \boldsymbol{x}} \text{OWA}_{\boldsymbol{w}}(\boldsymbol{C}\boldsymbol{x}) \right) \tag{27}$$

For the Moreau-envelope smoothed OWA optimization (13) proposed for end-to-end training, the main difference is that its objective function is differentiable (with gradients (32)), which allows solution by a more efficient Frank-Wolfe method Beck [2017], whose inner optimization over $\Delta$ reduces to the simple argmax function which returns a binary vector with unit value in the highest vector position and 0 elsewhere, which can be computed in linear time:

$$\boldsymbol{x}^{k+1} = \frac{k}{k+2} \boldsymbol{x}^k + \frac{2}{k+2} \text{argmax} \left( \frac{\partial}{\partial \boldsymbol{x}} \text{OWA}_{\boldsymbol{w}}(\boldsymbol{C}\boldsymbol{x}^k) \right) \tag{28}$$

# B  FAIR LEARNING TO RANK EXPERIMENT

## B.1  FAIR RANKING OPTIMIZATION BY FRANK-WOLFE WITH SMOOTHING

This section explains the adaptation of a Frank-Wolfe method with objective smoothing, due to Do and Usunier [2022], to solve the fair ranking optimization mapping (26) proposed for end-to-end fair learning to rank in this paper.

Frank-Wolfe methods solve a convex constrained optimization problem $\text{argmax}_{\mathbf{x} \in \mathbb{S}} f(\mathbf{x})$ by computing the iterations

$$\mathbf{x}^{(k+1)} = (1 - \alpha^{(k)})\mathbf{x}^{(k)} + \alpha^{(k)} \underset{\mathbf{y} \in \mathbb{S}}{\text{argmax}} \langle \mathbf{y}, \nabla f(\mathbf{x}^{(k)}) \rangle. \tag{29}$$

Convergence to an optimal solution is guaranteed when $f$ is *differentiable* and with $\alpha^{(k)} = \frac{2}{k+2}$ [Beck, 2017]. However, the main obstruction to solving (26) by the method (29) is that $f$ in our case includes a *non-differentiable* OWA function. A path forward is shown in [Lan, 2013], which shows convergence can be guaranteed by optimizing a smooth surrogate function $f^{(k)}$ in place of the nondifferentiable $f$ at each step of (29), in such a way that the $f^{(k)}$ converge to the true $f$ as $k \to \infty$.

It is proposed in [Do and Usunier, 2022] to solve a two-sided fair ranking optimization with OWA objective terms, by the method of [Lan, 2013], where $f^{(k)}$ is chosen to be a Moreau envelope $h^{\beta_k}$ of $f$, a $\frac{1}{\beta_k}$-smooth approximation of $f$ defined as [Beck, 2017]:

$$h^\beta(\mathbf{x}) = \min_{\mathbf{y}} f(\mathbf{y}) + \frac{1}{2\beta} \|\mathbf{y} - \mathbf{x}\|^2. \tag{30}$$

When $f = \text{OWA}_{\boldsymbol{w}}$, let its Moreau envelope be denoted $\nabla \text{OWA}_{\boldsymbol{w}}^\beta$; it is shown in [Do and Usunier, 2022] that its gradient can be computed as a projection onto the permutahedron induced by modified OWA weights $\tilde{\boldsymbol{w}} = -(w_m, \ldots, w_1)$. By definition, the permutahedron $\mathcal{C}(\tilde{\boldsymbol{w}}) = \text{CONV}(\{\boldsymbol{w}_\sigma : \forall \sigma \in \mathcal{P}_m\})$ induced by a vector $\tilde{\boldsymbol{w}}$ is the convex hull of all its permutations. In turn, it is shown in [Blondel et al., 2020] that the permutahedral projection $\nabla \text{OWA}_{\boldsymbol{w}}^\beta(\mathbf{x}) = \text{proj}_{\mathcal{C}(\tilde{\boldsymbol{w}})}(\boldsymbol{x}/\beta)$ can be computed in $m \log m$ time as the solution to an isotonic regression problem using the Pool Adjacent Violators algorithm. To find the overall gradient of $\text{OWA}_{\boldsymbol{w}}^\beta$ with respect to optimization variables $\boldsymbol{\Pi}$, a convenient form can be derived from the chain rule:

$$\nabla_{\boldsymbol{\Pi}} \text{OWA}_{\boldsymbol{w}}^\beta(\mathcal{E}(\boldsymbol{\Pi})) = \boldsymbol{\mu} \boldsymbol{b}^T. \tag{31}$$

where $\boldsymbol{\mu} = \text{proj}_{\mathcal{C}(\tilde{\boldsymbol{w}})}(\mathcal{E}(\boldsymbol{\Pi})/\beta)$ and $\mathcal{E}(\boldsymbol{\Pi})$ is the vector of all item exposures [Do and Usunier, 2022]. For the case where group exposures $\mathcal{E}_G(\boldsymbol{\Pi})\mathcal{E}_g(\boldsymbol{\Pi}) = \mathbf{1}_g^T \boldsymbol{\Pi} \boldsymbol{b}$ are aggregated by OWA, $\mathcal{E}_G(\boldsymbol{\Pi}) = \boldsymbol{A}\boldsymbol{\Pi}\boldsymbol{b}$, where $\boldsymbol{A}$ is the matrix composed of stacking together all group indicator vectors $\mathbf{1}_g \, \forall g \in G$. Since $\mathcal{E}(\boldsymbol{\Pi}) = \boldsymbol{\Pi}\boldsymbol{b}$, this implies $\mathcal{E}_G(\boldsymbol{\Pi}) = \mathcal{E}(\boldsymbol{A}\boldsymbol{\Pi})$, thus

$$\nabla_{\boldsymbol{\Pi}} \text{OWA}_{\boldsymbol{w}}^\beta(\mathcal{E}_G(\boldsymbol{\Pi})) = (\boldsymbol{A}^T \tilde{\boldsymbol{\mu}}) \, \boldsymbol{b}^T. \tag{32}$$

by the chain rule, and where $\tilde{\boldsymbol{\mu}} = \text{proj}_{\mathcal{C}(\tilde{\boldsymbol{w}})}(\mathcal{E}_G(\boldsymbol{A}\boldsymbol{\Pi})/\beta)$. It remains now to compute the gradient of the user relevance term $u(\boldsymbol{\Pi}, \hat{\boldsymbol{y}}_q) = \hat{\boldsymbol{y}}_q^T \boldsymbol{\Pi} \, \boldsymbol{b}$ in Problem 26. As a linear function of the matrix variable $\boldsymbol{\Pi}$, its gradient is $\nabla_{\boldsymbol{\Pi}} u(\boldsymbol{\Pi}, \hat{\boldsymbol{y}}_q) = \hat{\boldsymbol{y}}_q \, \boldsymbol{b}^T$, which is evident by comparing to the equivalent vectorized form $\hat{\boldsymbol{y}}_q^T \boldsymbol{\Pi} \, \boldsymbol{b} = \overrightarrow{\hat{\boldsymbol{y}}_q \, \boldsymbol{b}^T} \cdot \overrightarrow{\boldsymbol{\Pi}}$. Combining this with (32), the total

**Algorithm 1:** Frank-Wolfe with Moreau Envelope Smoothing to solve (26)

---

**Input:** predicted relevance scores $\hat{\boldsymbol{y}} \in \mathbb{R}^n$, group mask $\boldsymbol{A}$, max iteration T, smooth seq. $(\beta_k)$
**Output:** ranking policy $\boldsymbol{\Pi}^{(T)} \in \mathbb{R}^{n \times n}$

1 Initialize $\boldsymbol{\Pi}^{(0)}$ as $\boldsymbol{P} \in \mathcal{P}$ which sorts $\hat{\boldsymbol{y}}$ in decreasing order;
2 **for** $k = 1, \ldots, T$ **do**
3     $\tilde{\boldsymbol{\mu}} \leftarrow \text{proj}_{\mathcal{C}(\tilde{\boldsymbol{w}})}(\mathcal{E}_G(\boldsymbol{A}\boldsymbol{\Pi})/\beta_k)$;
4     $\hat{\boldsymbol{\mu}} \leftarrow (1 - \lambda) \cdot \hat{\boldsymbol{y}}_q + \lambda \cdot (\boldsymbol{A}^T \tilde{\boldsymbol{\mu}})$;
5     $\hat{\sigma} \leftarrow argsort(-\hat{\boldsymbol{\mu}})$;
6     Let $\boldsymbol{P}^{(k)} \in \mathcal{P}$ such that $\boldsymbol{P}^{(k)}$ represents $\hat{\sigma}$;
7     $\boldsymbol{\Pi}^{(k)} \leftarrow \frac{k}{k+2}\boldsymbol{\Pi}^{(k-1)} + \frac{2}{k+2}\boldsymbol{P}^{(k)}$;
8 Return $\boldsymbol{\Pi}^{(T)}$;

---

gradient of the objective function in (26) with smoothed OWA term is $(1 - \lambda) \cdot \hat{\boldsymbol{y}}_q\ \boldsymbol{b}^T + \lambda \cdot (\boldsymbol{A}^T \tilde{\boldsymbol{\mu}})\ \boldsymbol{b}^T$, which is equal to $\left((1 - \lambda) \cdot \hat{\boldsymbol{y}}_q + \lambda \cdot (\boldsymbol{A}^T \tilde{\boldsymbol{\mu}})\right)\ \boldsymbol{b}^T$. Therefore the SOFaiR module's Frank-Wolfe linearized subproblem is

$$\underset{\boldsymbol{\Pi} \in \mathcal{B}}{\arg\max} \left\langle \boldsymbol{\Pi}, \left((1 - \lambda) \cdot \hat{\boldsymbol{y}}_q + \lambda \cdot (\boldsymbol{A}^T \tilde{\boldsymbol{\mu}})\right)\ \boldsymbol{b}^T \right\rangle \tag{33}$$

To implement the Frank-Wolfe iteration (29), this linearized subproblem should have an efficient solution. To this end, the form of each gradient above as a cross-product of some vector with the position biases $\boldsymbol{b}$ can be exploited. Note that as the expected DCG under relevance scores $\boldsymbol{y}$, the function $\boldsymbol{y}^T \boldsymbol{\Pi}\ \boldsymbol{b}$ is maximized by the permutation matrix $\boldsymbol{P} \in \mathcal{P}_n$ which sorts the relevance scores $\boldsymbol{y}$ decreasingly. But since $\boldsymbol{y}^T \boldsymbol{\Pi}\ \boldsymbol{b} = \overrightarrow{\boldsymbol{y}\ \boldsymbol{b}^T} \cdot \overrightarrow{\boldsymbol{\Pi}}$, we identify $\boldsymbol{y}^T \boldsymbol{\Pi}\ \boldsymbol{b}$ as the linear function of $\overrightarrow{\boldsymbol{\Pi}}$ with gradient $\overrightarrow{\boldsymbol{y}\ \boldsymbol{b}^T}$. Therefore problem (33) can be solved in $\mathcal{O}(n \log n)$, simply by finding $\boldsymbol{P} \in \mathcal{P}_n$ as the argsort of the vector $\left((1 - \lambda) \cdot \hat{\boldsymbol{y}}_q + \lambda \cdot (\boldsymbol{A}^T \tilde{\boldsymbol{\mu}})\right)$ in decreasing order. A more formal proof, cited in [Do et al., 2021], makes use of [Hardy et al., 1952].

### B.2 RUNNING TIME ANALYSIS

Our analysis begins with a runtime comparison between SOFaiR and other LTR frameworks, to show how it overcomes inefficiency at training and inference time. Figure 8 shows the average training and inference time per query for each method, focusing on the binary group MSLR dataset across various list sizes. First notice the drastic runtime reduction of SOFaiR compared to SPOFR, both during training and inference. While SPOFR's training time exponentially increases with the ranking list size, SOFaiR's runtime increases only moderately, reaching over one order of magnitude speedup over SPOFR for large list sizes. Notably, the number of iterations of Algorithm 1 required for sufficient accuracy in training to compute SPO+ subgradients are found to less than those required for solution of (26) at inference. Thus the reported results use 100 iterations in training and 500 at inference. Importantly, reported runtimes under-estimate the efficiency gained by SOFaiR, since its PyTorch [Paszke et al., 2017] implementation in Python is compared against the highly optimized code implementation of Google OR-Tools solver [Perron, 2011]. DELTR and FULTR, being penalty-based methods, demonstrate competitive runtime performance. However, this efficiency comes at the expense of their ability to ensure fairness in every generated policy.

### B.3 MODELS AND HYPERPARAMETERS

**Models and hyperparameters.** A neural network (NN) with three hidden layers is trained using Adam Optimizer with a learning rate of 0.1 and a batch size of 256. The size of each layer is halved, and the output is a scalar item score. Results of each hyperparameter setting is are taken on average over five random seeds.

Fairness parameters, considered as hyperparameters, are treated differently. LTR systems aim to offer a trade-off between utility and group fairness, since the cost of increased fairness results in decreased utility. In DELTR, FULTR, and SOFaiR, this trade-off is indirectly controlled through the fairness weight, denoted as $\lambda$ in (26). Larger values of $\lambda$ indicate more preference towards fairness. In SPOFR, the allowed violation of group fairness is specified directly. Ranking utility and fairness violation are assessed using average DCG and fairness violation, respectively. The metrics are computed as averages over the entire test dataset.

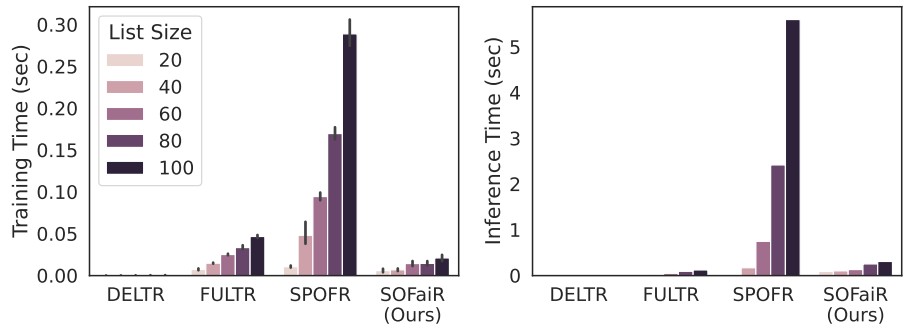

Figure 8: Running time benchmark on MSLR-Web10k dataset

Table 2: Hyperparameters

| Hyperparameter | Min | Max | Final Value | | | |
|---|---|---|---|---|---|---|
| | | | SOFaiR | SPOFR | FULTR | DELTR |
| learning rate | $1e^{-5}$ | $1e^{-1}$ | $\mathbf{1e^{-1}}$ | $\mathbf{1e^{-1}}$ | $\mathbf{2.5e^{-4}}$ | $\mathbf{2.5e^{-4}}$ |
| violation penalty $\lambda$ | $1e^{-5}$ | 400 | * | N/A | * | * |
| allowed violation $\delta$ | 0 | 0.01 | N/A | * | N/A | N/A |
| entropy regularization decay | 0.1 | 0.5 | N/A | N/A | 0.3 | N/A |
| batch size | 64 | 512 | 256 | 256 | 256 | 256 |
| smoothing parameter $\beta_0$ | 0.1 | 100 | * | N/A | N/A | N/A |
| sample size | 32 | 64 | N/A | 64 | 64 | N/A |

Hyperparameters were selected as the best-performing on average among those listed in Table 1. Final hyperparameters for each model are as stated also in Table 2, and Adam optimizer is used in the production of each result. Asterisks (*) indicate that there is no option for a final value, as all values of each parameter are of interest in the analysis of fairness-utility tradeoff, as reported in the experimental setting Section.

For OWA optimization layers, $\mathbf{w}$ is set as $w_j = \frac{n-1+j}{n}$, $T = 100$ during training , and $T = 500$ during testing.

## B.4 ADDITIONAL RESULTS

This section includes additional results for fair learning to rank on MSLR, in which list sizes to be ranked are increased to 100 items. This allows runtimes to be compared as a function of list size, which determines the size of the fair ranking optimization problem. It also reveals how penalty-based methods DELTR and FULTR suffer in their ability to satisfy fairness accurately.

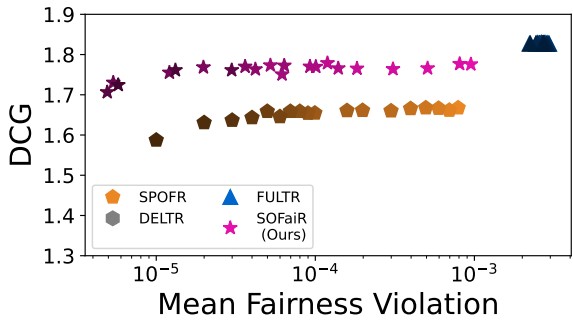

Figure 9: Fairness-utility tradeoffs on MSLR

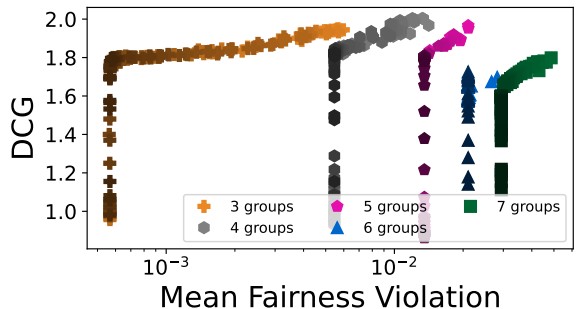

Figure 10: Fairness-utility tradeoffs on MSLR Multi-group

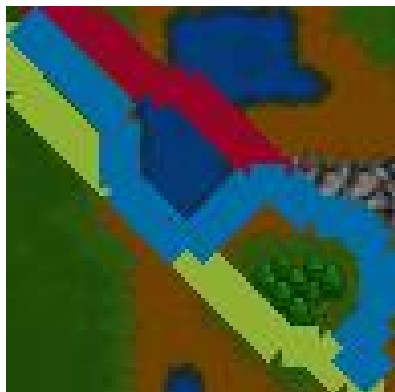

Figure 11: Shortest paths for each species in a Warcraft map

| Model | Human | | | Naga | | | Dwarf | | |
|---|---|---|---|---|---|---|---|---|---|
| | 25 | 50 | 100 | 25 | 50 | 100 | 25 | 50 | 100 |
| Two-Stage MSE Loss | 44.4 | 44.6 | 46.3 | **34.2** | **34.1** | **33.9** | 44.1 | **41.6** | **42.8** |
| End2End Sum Loss | 51.5 | 49.0 | 47.9 | 35.2 | 33.6 | 34.6 | 43.8 | 43.8 | 43.4 |
| **End2End OWA Loss** | **43.8** | **31.8** | **33.6** | **34.2** | 37.6 | 34.8 | **41.3** | 43.1 | 43.1 |

Table 3: Regret (%) per species

## C   MULTI-SPECIES WARCRAFT SHORTEST PATH

Figure 11 showcases the Warcraft map featuring the shortest paths for three distinct species. The paths for Humans, Naga, and Dwarves are depicted in green, red, and blue, respectively. Humans excel on land, Naga traverse water most efficiently, while Dwarves navigate rocky terrain with the greatest speed.

Table 3 presents the regrets for each species across three models with different number of training data. It is notable that the model trained with OWA Loss significantly outperforms the two-stage model by more than 10% for the Human race. Conversely, the two-stage model exhibits slightly better performance for the Dwarf, albeit by a very small margin (<3%).

