# OpenReview forum: "End-to-End Learning for Fair Multiobjective Optimization Under Uncertainty"
_auai.org/UAI/2024/Conference — UAI 2024 poster_

### Official Review · Reviewer_F8Fc · 2024-03-18

**Q2-1 Originality-Novelty:** 2
**Q2-2 Correctness-Technical Quality:** 3
**Q2-5 Clarity Of Writing:** 2

**Q1 Summary And Contributions:**

The authors extend the predict-then-optimize paradigm from single to multiple objectives under constraints. They do so by concatenating the (vector valued) objective function with a permuted weighted sum (the OWA function), claimed to result in a “fair” and robust aggregation of the objectives. Since OWA is not a smooth function, hence, direct application of gradient-based optimization methods tend to fail, the authors provide two smoothing approaches of the OWA in order to solve this issue. The is applicable in case the constraints and objective function are linear. Then, the problem can be translated into a linear program where solutions are well known. The major drawback is the computational effort. The second approach is based on the Moreau smoothing technique. In this setting, the original (non-smooth) function is replaced by the Moreau envelope function which (under certain assumptions) is smooth and shares the same optima.

**Q2-3 Extent To Which Claims Are Supported By Evidence:**

3: Good: the main claims are supported by convincing evidence (in the form of adequate experimental evaluation, proofs, (pseudo-)code, references, assumptions).

**Q2-4 Reproducibility:**

3: Good: key resources (e.g. proofs, code, data) are available and key details (e.g. proofs, experimental setup) are sufficiently well-described for competent researchers to confidently reproduce the main results.

**Q3 Main Strengths:**

The generalization of the predict-then-optimize framework to the multi-objective paradigm is an interesting and novel idea.

**Q4 Main Weakness:**

**Motivation of OWA citerion**

I have difficulties understanding the choice of OWA in the first place:

* It may be worthwhile to mention that the original paper proposing OWA (Ogrycacz et al) does not claim the OWA aggregation function to be “fair”. Other sources (e.g. Do et al.) consider special setups, such as recommendation systems, in which it is more reasonable to claim the OWA to be fair. Unfortunately, the paper does not provide additional motivation for calling the method "fair".

* The OWA is claimed to be “fair” in the sense of being invariant under permutation of the objectives. However, if the ranges of the objectives don't overlap (which frequently occurs in practice), the OWA’s underlying sorting becomes constant. In other words, the OWA equals a standard weighted sum. In particular, OWA heavily depends on the scaling of the objectives. In contrast, I would expect a “fair” optimization for general settings to be invariant under the scale of the objectives.

* The OWA’s underlying sorting is the cause of non-differentiability of the aggregation and the main justification of the paper’s central methodology (i.e. OWA-QP and OWA-Moreau). If the OWA is replaced by a standard weighted sum for example, the central problem becomes much simpler since the aggregation becomes differentiable. Why is it imperative to look at such a difficult objective in the first place?

**Clarity of writing**

I found the paper quite difficult to read, mainly because of a lack of mathematical precision in many places. See the detailed comments for examples.

**Q5 Detailed Comments To The Authors:**

**Mathematical study**

I encourage the authors to make some effort in properly expressing and justifying their mathematical results.: In addition to Q4 there are several minor mathematical errors, to name a few:

* in section 2.1 the weights w_i need to be strictly greater than 0 rather than being greater or equal zero in order for the weighted sum approach to yield Pareto optimal solutions. This is also recognized in the respective reference (Ogrycacz et al).
* equation (7)  should yield a set (of subgradients) rather than a single vector
* the moreau function is defined as an infimum rather than a minimum (equation (12)); It may be reasonable to justify why this infimum exists.
* there are inconsistencies within the notation (for example equation (8) uses the transposed symbol while it is neglected in equations (9) (10c) etc)

**Discussion of empirical study**

The description of the empirical study needs more detail:

* What are methods OWA-LP, Sum-QP exactly?
* Why is the regret given in percent? Percentage of what?
* Can you explain why two-stage OWA is so much worse?


**Post-rebuttal update**

Since the my main technical concerns have been addressed, I raise my score to a 6.

**Q9 Complying With Reviewing Instructions:**

Yes

---

> ### Author Rebuttal · Authors · 2024-04-05
>
> Thanks for the detailed feedback in this review. We are happy to address all the reviewer’s concerns below.
>
> First please note the following factual inaccuracy:
>
> > **“The authors do not provide code for their experiments.”**
>
> Our full source code was submitted as supplementary material in the original submission. This is easy to verify.
>
>
> Below, we respond to the review's bulleted comments in their respective order, organized by each section of the review.
>
> ### Mathematical study
>
> The review alleges “several minor mathematical errors”, of which four are listed. Three out of four of those alleged errors are not actually errors. Below we respond to each, in the order listed in the review. We also invite the review to share the other perceived inaccuracies not listed if any, so that we have a chance to respond to them.
>
>
> - **“The weights need to be strictly positive…”** This is correct to point out, thanks for noticing! Note that this is a technical edge case, as zero OWA weights would defeat the purpose of using Fair OWA, by ignoring some criteria. This point will be revised by adding the condition $w_n>0$ for Fair OWA in Section 2.1.
>
> - **“Eq (7) should yield a set of sub gradients…”** We characterize the formula as one which produces a single subgradient and not the set of all possible subgradients. It is not necessary to find the set of all subgradients; a single subgradient is sufficient to enable optimization by subgradient descent.
>
> - **“The Moreau function is defined as an infimum rather than a minimum”.** It’s important to note that the Moreau envelope is not always defined as an infimum (as opposed to a minimum). See for example reference [1], a standard graduate-level textbook on optimization. They define the Moreau envelope in Section 6.7 as a minimum, which agrees exactly with our definition. Reference [2] is a related research paper that also shares our exact definition of the Moreau envelope. We acknowledge that the Moreau envelope is sometimes defined using the infimum, for example in reference [4]. The notions of “min” and “inf” are often considered interchangeable in optimization theory, where functions are assumed to be closed, proper, and convex (concave). Of course, our OWA functions fit these criteria.
>
> - We are happy to change the inner product in (9) and (10c) to use the transpose notation if it aids in clarity. However, it is not a mathematical error. Is completely correct with respect to universally adopted notation.
>
>
> > **“Questionable validity of Moreau envelope method”**
>
> The review cites this as the paper’s main weakness, implying that the paper contains a methodological flaw stemming from incorrect use of the Moreau envelope. This feedback is misleading and misinterprets the paper.
>
> The review is correct to point out that our definition of the Moreau envelope is valid specifically for convex functions, while our OWA objective is concave rather than convex. This was a conscious decision, as the notions of convexity and concavity are equivalent to a minus sign. If f is concave, then -f is convex. Also, to minimize f is to maximize -f. These equivalences are considered trivialities in optimization and are often applied without explicit explanation. Note for example that reference [2] also defines the Moreau envelope for convex functions, to preface a method that instead maximizes a concave function. Our definitions including equation (12) are meant to match common conventions in optimization, where definitions are typically oriented for minimization problems.
>
> The reviewer’s suggested adjustment is of course correct and we thought it would be considered obvious, but they pose it as a solution to a technical flaw in the paper, which seems strange. Please note that we welcome suggestions as to where such ambiguities are better explicitly clarified in the paper. We rely on the reviewers’ feedback to help us understand which conventions are most appropriate for the UAI audience. As such, we are happy to redefine the Moreau envelope for concave functions in the paper if it would improve the clarity. However, this review frames our expositional choice as a methodological flaw. There is no such flaw. That being said, we emphasize that **the reviewer’s technical remarks on the “Questionable validity of the Moreau envelope method” are not correct**. Our Moreau envelope is in fact $1/(\beta)$ smooth by construction, and it does share the same optima of the underlying OWA objective function. Our empirical results also attest to the methods’ soundness and correct implementation.

---

### Official Review · Reviewer_3Qx8 · 2024-03-21

**Q2-1 Originality-Novelty:** 2
**Q2-2 Correctness-Technical Quality:** 3
**Q2-5 Clarity Of Writing:** 2

**Q1 Summary And Contributions:**

The paper is interested on predict-then-optimize framework when the optimization objective is a AWO aggregation. More precisely they focus on Fair OWA when the weights have decreasing order.  In fact, in this case the OWA optimization can be linearized. The prediction focus on the cost coefficient for fixed OWA weights.
The paper proposes a differentiable approximation of OWA based on the linear formulation of fair OWA by regularization of smooth functions. Then a smoothing via Moreau envelope is proposed.
Then a black box method focusses on the case of convex aggregation of linear function and OWA. Then, a section discusses some limitations.
In the experimental section all the clams are illustrated.

**Q2-3 Extent To Which Claims Are Supported By Evidence:**

2: Fair: the main claims are somewhat supported by evidence (but the experimental evaluation may be weak, or does not match entirely with the claims, important baselines may be missing, proofs contain important ideas but lack rigor, algorithmic details are only discussed superficially, references are imprecise, assumptions are not sufficiently motivated or explicated, etc.).

**Q2-4 Reproducibility:**

2: Fair: key resources (e.g. proofs, code, data) are unavailable but key details (e.g. proof sketches, experimental setup) are sufficiently well-described for an expert to confidently reproduce the main results.

**Q3 Main Strengths:**

The proposition of Moreau envelope smoothing is interesting and the experimental results show the effectiveness of the approach.

**Q4 Main Weakness:**

The author used the exponential LP formulation of OWA based on paper Ogryczak and Sliwisnki 2003 while in the same paper the authors propose a polynomial formulation and show that the compact formulation is better.

**Q5 Detailed Comments To The Authors:**

Section 4.1
I do not understand why if it is the case of fair OWA the number of constraints grow factorially in model (10). In Ogryczak and Sliwisnki 2003 a model with polynomial number of constraint is proposed model (30-34).  Why do not use this one?
Section 5
The argument to propose an alternative is that the constraint growing factorially. But since they exist a LP with polynomial number of constraints There is other arguments to propose the black box model?
Section 6 the “surrogate approach refer to “Moreau envelope smoothing”  or to the “blackbox methods”? If it is about “Moreau envelope smoothing” it is better to explain before in section 5 the limitation of the approach.
Section 7. OWA-QP better perform than OWA-Moreau the weakness is du to the exponential formulation of OWA which can be easy transform to polynomial formulation. So the interest of the Moreau envelope smoothing is still not clear.

**Q9 Complying With Reviewing Instructions:**

Yes

---

> ### Author Rebuttal · Authors · 2024-04-05
>
> We thank Reviewer 3Qx8 for their review and comments, and for acknowledging that **the proposition of Moreau envelope smoothing is interesting and the experimental results show the effectiveness of the approach**.
>
> The reviewer states that Ogryczak 2002 (see [2]) also proposes a polynomial-sized alternative LP formulation of the OWA optimization, and that this should be easy to incorporate in our framework rather than the proposed OWA-Moreau method.
>
> These concerns are easily addressed. Firstly, note that our ultimate method in this problem setting is OWA-Moreau, not OWA-QP, as indicated in the paper. OWA-QP is shown to be effective in end-to-end learning, but not scalable (see Section 4 overview). The defining advantage of OWA-Moreau is its scalability.
>
> With regards to the reviewer's suggestion, it is not true that their alternative polynomial-sized formulation is easy to incorporate in our framework, both from an efficiency point of view and a modeling point of view.
>
> From a modeling POV: The factorial-sized OWA LP used in our paper requires no change of variables (it considers objective values as additional variables) and is based mainly on adding constraints. In contrast, the polynomial-sized OWA LP requires a change of variables which must also be reversed after solving in order to recover solutions to the underlying OWA problem (see [2]). However, the differentiability of such optimization requires smoothing of its objective function, which changes the problem. There is no reason to believe that applying the reverse change of variables to the solution of a problem modified by such smoothing should return an accurate solution to the underlying OWA optimization. It’s not even clear that it would provide a feasible solution to the underlying OWA problem in its original variables.
>
> From an efficiency POV: It's important to emphasize that in any case, we would not be solving the linear programs proposed in Ogryczak 2002. Instead, we would be solving Quadratic Programs derived from augmenting their objective functions with smooth quadratic terms. These QP's are much harder to solve than their associated LP problems. Regardless, despite scaling better than a factorial, the polynomial-sized OWA LP is itself still not very scalable. Its number of variables and constraints both increase quadratically with the size of the underlying OWA problem. In backpropagation, the resulting linear system for the Jacobian of its smoothed QP approximation (see Section 3) will also scale poorly, since it is based on differentiating the KKT conditions (see [1]), in which the number of primal and dual variables both grow quadratically.
>
> We hope these points can be discussed among all the expert reviewers and the AC, since they are important.
>
> Our OWA-Moreau approach is meant to be a truly scalable alternative to the quadratic smoothing approach. The per-iteration cost of its gradient-based optimization scales as $m log m$ (where $m$ is the number of criteria) and its Jacobian in backpropagation can be found by solving a linear system of size equal to the number of variables in the underlying OWA problem.

---

### Official Review · Reviewer_DrsE · 2024-03-23

**Q2-1 Originality-Novelty:** 3
**Q2-2 Correctness-Technical Quality:** 3
**Q2-5 Clarity Of Writing:** 3

**Q1 Summary And Contributions:**

The paper proposes several techniques (i.e., via subgradients, quadratic smoothing, or Moreau envelope smoothing) to optimize a fair OWA criterion in the predict-then-optimize framework. Various variations of this framework are also discussed (e.g., linear surrogate with integer constraints). The approaches are validated on several problems (i.e., portfolio optimization, shortest path, learning-to-rank).

**Q2-3 Extent To Which Claims Are Supported By Evidence:**

3: Good: the main claims are supported by convincing evidence (in the form of adequate experimental evaluation, proofs, (pseudo-)code, references, assumptions).

**Q2-4 Reproducibility:**

2: Fair: key resources (e.g. proofs, code, data) are unavailable but key details (e.g. proof sketches, experimental setup) are sufficiently well-described for an expert to confidently reproduce the main results.

**Q3 Main Strengths:**

The investigated framework and the proposed techniques seem to be quite generic, although the presentation focuses on linear programs. At least, the Moreau envelope smoothing approach is quite new in this context, as far as I know.

Sufficient experiments are conducted to understand how the different approaches compare.

The best paper is overall relatively well-written, although some statements are incorrect (see below).

**Q4 Main Weakness:**

Some sentences should be rephrased:
- Below (2): the general OWA is not concave. Only fair OWA is.
- Below (7): Subgradient (7) has already been used in the training of machine learning, e.g.,
Siddique et al., Learning Fair Policies in Multi-Objective (Deep) Reinforcement Learning with Average and Discounted Rewards, ICML, 2020

Unless I've missed it, I think it would be good:
- to discuss the extension to non-linear programs. What would be the difficulties?
- to provide some intuitive explanation of why the different propositions work well (or not).
- to mention if the results would be sensitive to other choices of OWA weights.

**Q5 Detailed Comments To The Authors:**

See also above.

Could the reinforcement learning problem studied in Siddique et al. also fit in some ways the predict-then-optimize framework?

When a citation is not part of a sentence, it should be between brackets.

**Q9 Complying With Reviewing Instructions:**

Yes

---

> ### Author Rebuttal · Authors · 2024-04-05
>
> We thank Reviewer DrsE for their helpful review and comments. In particular thank you for acknowledging the novelty of the **Moreau envelope smoothing approach**, its clarity and experimental setup.
>
> We provide below our answers to your questions and comments.
>
> > **“Only Fair OWA is concave…”:**
>
> Thanks for catching this misphrasing! It’s revised now. Greatly appreciate it.
>
> > **“Subgradient (7) has been used...”**
>
> Thanks, we were not able to find this in our literature search. It’s quite relevant and will be mentioned and cited in the final version!
>
> > **"Unless I've missed it, I think it would be good..."**
>
> We agree with the suggestions provided and will make efforts to work them in. It’s helpful to know that the exclusion of nonlinear objectives is noticeable. Restricting to the linear case helps simplify the exposition and covers our main applications of interest. But we could benefit from some discussion at the end. In short, most of the proposed methods should carry over to the nonlinear case, possibly barring cases where that nonlinearity would cause the overall OWA aggregation to become nonconvex.
>
> We'll also make efforts to better motivate the intuitions behind some of the PtO methods, for readers less familiar with the topic.
>
> > **“Could the reinforcement learning problem studied in Siddique et al. also fit in some ways the predict-then-optimize framework?”**
>
> Thank you for sharing this reference. There are some similarities  in that we train to optimize an OWA objective in order to make fair optimal decisions over multiple criteria. However, our work differs in that it incorporates an OWA optimization model directly the training loop. The main advantage of this proposal is guaranteed satisfaction of the constraints. Fundamentally we model decisions as feasible solutions within a constrained set. In Siddique, it would seem that they instead model decisions as a sequence of choices. That setting is probably better treated with the RL / policy gradient approach. We might conjecture that in the vain of Bello ’17, Siddique may be well-suited to learning to solve combinatorial OWA problems, as combinatorial solutions are sometime modeled as sequences, as long as they don't require complex constraints. We will make sure to cite this work in our final version.
>
>
> We appreciate your helpful feedback and questions. Please let us know if there are any further comments or questions, we are happy to discuss.

---

### Official Review · Reviewer_3fJX · 2024-03-23

**Q2-1 Originality-Novelty:** 3
**Q2-2 Correctness-Technical Quality:** 3
**Q2-5 Clarity Of Writing:** 3

**Q10 Ethical Concerns:**

No.

**Q1 Summary And Contributions:**

The paper extends the Predict Then Optimize methodology to problems where the objectives are Ordered Weighted Averaging. This requires to solve the issue of the non-differentiability, that prevents backpropagation of the constrained optimization mapping within machine learning models trained by gradient descent. The paper proposes a few techniques for differentiating OWA optimization models, and puts forward some strategies for combining parametric prediction with OWA optimization.

**Q2-3 Extent To Which Claims Are Supported By Evidence:**

3: Good: the main claims are supported by convincing evidence (in the form of adequate experimental evaluation, proofs, (pseudo-)code, references, assumptions).

**Q2-4 Reproducibility:**

2: Fair: key resources (e.g. proofs, code, data) are unavailable but key details (e.g. proof sketches, experimental setup) are sufficiently well-described for an expert to confidently reproduce the main results.

**Q3 Main Strengths:**

The use of OWAs within the PtO methodology is, to the best of my knowledge, new. The paper manages to make use of the advantages provided by the OWAs while being able to overcome their weaknesses. It is well-written and fits within the scope of the conference.  The experimental part is detailed and sound.

**Q4 Main Weakness:**

The technical improvent of the paper is a bit incremental, given that it builds upon existing techniques from other papers.

**Q5 Detailed Comments To The Authors:**

The paper presents a fair case of the challenges and advantages related to the use of the OWAs. I have no major objections. Still, it would perhaps be interesting to expand a bit the ideas of the OWAs presented in section 2.1, and to discuss in more detail while they are an interesting alternative to other approaches; this more so given that, as discussed by the authors in section 8, there are also many related works facing similar challenges.

**Q9 Complying With Reviewing Instructions:**

Yes

---

> ### Author Rebuttal · Authors · 2024-04-05
>
> We thank Reviewer 3fJX for their helpful review, in particular their acknowledgement of the novelty of our work and about its ability to **use of the advantages provided by the OWAs while being able to overcome their weaknesses**, while being **well-written** and have **detailed and sound experimental part**.
>
> We provide below our responses to your helpful comments and questions.
>
> > **“The technical improvement of the paper is a bit incremental, given that it builds upon existing techniques from other papers.”**
>
> We’d like to make a few points that may help clarify the value of the paper’s contributions:
>
> 1. Firstly, the paper makes significant methodological contributions. We choose to focus on OWA optimization, but some of the techniques proposed are generic and may be applied to other problem settings.
>
> The reason is that OWA, as a nondifferentiable objective, poses novel challenges to differentiable optimization in PtO. **We’re not aware of any other PtO models where the objective itself is non-differentiable** and this challenge leads us to be **the first to propose smoothing via the Moreau envelope** as a natural solution to overcome it (Section 4.2). That idea seems likely to generalize to other cases in PtO in which non-differentiable objectives are required. To our knowledge, **the idea of using an LP problem to maintain integer constraints in the loop via total unimodularity (Section 7.2.1)** is also new and could be useful elsewhere.
>
> 2. The paper also solves significant technical challenges in interfacing methodologies from different domains. Despite that it leverages techniques from classical optimization and modern works (most papers in the integration of ML and optimization inevitably do so), they must be combined and modified in very nontrivial ways, to take full advantage of OWA properties without succumbing to their disadvantages.
> Using the LP formulations of OWA as a bridge to PtO techniques for LP, and using a differentiable implementation of the OWA's Moreau envelope gradient to implicitly differentiate the overall OWA optimization, are nontrivial design choices that require a deep toolkit. The full logic behind those choices may not come across clearly in the paper as presented, where we only have space to focus on what ultimately works best. We'll plan to make an account of why many existing PtO techniques cannot be employed in our setting, in the revised Appendix to better communicate the intuitions and lessons learned.
>
> 3. The experimental applications are novel and practically significant. They show that OWA’s use in PtO is worthy of the broader community’s attention. For example, while robust portfolio optimization is a common setting for studying OWA optimization, real-world portfolio modeling always has a predictive component and we provide the methodology to bridge that gap with end-to-end learning. We even demonstrate the methodology by using it to build new a fair learning-to-rank system, which shows significant advantages over existing methods in that space.
>
> 4. Finally, the paper is comprehensive within its scope. The paper may be limited to OWA in PtO, but it does a comprehensive job of (1) Contributing novel modeling techniques where appropriate  (2) Showing how existing PtO techniques can be adapted in nontrivial ways to serve best in this setting, and  (3) Building novel experimental applications which show the potential of OWA in PtO.
>
> > **“The paper is well-written but its interest may perhaps be better justified”**
>
> In response to the general concern that the incorporation of OWA in PtO should be better motivated, please see our included remarks above in the general response. OWAs are fundamental to modeling fairness in decision-making and therefore are of considerable interest in AI. In particular, joint models of prediction and decision-making often must be integrated in AI [1]. The paper makes substantial contributions towards enabling AI systems to learn fair decision-making. Your comment is helpful and we will make sure to highlight these aspects further in the final version of the manuscript!
>
> We appreciate the reviewer's feedback and helpful comments, and hope our responses have addressed your concerns so far. We are happy to answer any more questions, and are interested to know what if any improvements or clarifications would be needed for your full support of the paper.
>
> Citations:
> [1]: J. Kotary et al., "End-to-End Constrained Optimization Learning: A Survey", Proceedings of the International Joint Conference on Artificial Intelligence, 2021

---

### Meta-Review · Area_Chair_N5dd · 2024-04-17

The paper gives a concrete extension of the predict-and-optimize framework with a fairness criterion and experimental results are sufficiently convincing. For the cons, this is an incremental work based on an existing framework (still the Moreau envelope smoothing approach is new in this context) and there is a lack of intuitive explanations of the different contributions.